# Iodine Bioavailability and Accumulation of Arsenic and Cadmium in Rats Fed Sugar Kelp (*Saccharina latissima*)

**DOI:** 10.3390/foods11243943

**Published:** 2022-12-07

**Authors:** Even Fjære, Rikke Poulsen, Arne Duinker, Bjørn Liaset, Martin Hansen, Lise Madsen, Lene Secher Myrmel

**Affiliations:** 1Institute of Marine Research, NO-5817 Bergen, Norway; 2Environmental Metabolomics Laboratory, Department of Environmental Science, Aarhus University, DK-4000 Roskilde, Denmark; 3Laboratory of Genomics and Molecular Biomedicine, Department of Biology, University of Copenhagen, DK-2100 Copenhagen, Denmark

**Keywords:** seaweed, brown algae, sugar kelp, iodine, bioavailability, arsenic, cadmium, thyroid hormones, metabolomics, rats

## Abstract

Suboptimal iodine status is a prominent public health issue in several European coun-tries. Brown algae have a high iodine content that, upon intake, may exceed the recommended dietary intake level, but iodine bioavailability has been reported to be lower than from potassium iodide (KI) and highly depends on algae species. Further, potential negative effects from other components in algae, such as cadmium (Cd) and arsenic (As), have also been addressed. In this study, we observed a lower bioavailability of iodine from farmed sugar kelp (*Saccharina latissima*) than from KI in female Wistar IGS rats. Urinary iodine excretion was 94–95% in rats fed KI and 73–81% in rats fed sugar kelp, followed by increased faecal iodine levels in rats fed sugar kelp. No effects on body weight, feed efficiency, or plasma markers for liver or kidney damage were detected. The highest dose of iodine reduced plasma free thyroxine (fT4) and total T4 levels, but no significant effects on circulating levels of thyroid-stimulating hormone (TSH) and free triiodo-thyronine (fT3) were detected. Faeces and urine measurements indicate that 60–80% of total As and 93% of Cd ingested were excreted in rats fed 0.5 and 5% kelp. Liver metabolomic profiling demonstrates that a high inclusion of sugar kelp in the diet for 13 weeks of feeding modulates metabolites with potential antioxidant activity and phytosterols.

## 1. Introduction

The trace element iodine is essential for humans. Suboptimal iodine status and iodine deficiency are major problems of public health, and globally, approximately two billion people suffer from iodine deficiency, whereas 50 million exhibit clinical manifestations of iodine deficiency [1]. Children and breastfeeding women are the most vulnerable groups because the developing foetal brain and nervous system depend on maternal and foetal thyroid hormones. Thyroid-stimulating hormone (TSH) regulates triiodothyronine (T3) and thyroxine (T4) production, whereas iodine is necessary for T3 and T4 synthesis. Dietary iodine deficiency increases TSH secretion and induces thyroid hypertrophy. Additionally, a prolonged dietary iodine intake lower than 50 μg/day usually results in goiter. The recommended dietary iodine intake is 150 μg/day for adults [2]. Dairy products and seafood contain iodine, but dietary sources of iodine are limited. Therefore, iodised salt programs are recommended or mandatory in several countries [3,4]. Iodine supplementation and products added iodised salt have reduced the number of people at risk of iodine deficiency, but suboptimal iodine status remains evident in Europe [5,6]. Unlike most other parts of the world, Japan is considered an iodine-replete area because of the regular intake of seaweed in the diet [7].

The use of iodine supplements and products added iodised salt also caused concerns about excessive iodine exposure in some individuals or groups. The upper tolerable intake of iodine is set to 600 µg/day in Europe, 1100 µg/day in the USA, and 3000 µg/day in Japan [8]. Excessive iodine intake can cause thyroid dysfunction in susceptible individuals but is reported to generally be well tolerated. Most individuals tolerate a chronic iodide surplus of 30 mg up to 2 g per day. This induces persistent reduction in thyroid hormones T3 and T4 (15 and 25% reduction, respectively) and a rise in TSH without clinical symptoms [9]. However, excess iodine may cause clinical hypothyroidism in many vulnerable individuals, such as after treatment of Graves’ disease with radioiodine, after partial thyroidectomy, or in the presence of autoimmune thyroiditis. In subjects with dysfunctional TSH regulation of thyroid production (autonomous nodules), even normal dietary iodine levels may cause iodine-induced hyperthyroidism [9].

Different types of seaweed vary widely in iodine content, ranging from 16–8165 mg/kg dry weight [10], although, most brown algae have a very high iodine content. Regular consumption of seaweed, especially brown algae, could induce excessive iodine intake and possibly thyroid dysfunction. Kelp are large brown algae and generally a very rich source of iodine. For some kelp species, the consumption of one gram would exceed the tolerable upper intake limit for iodine. In particular, in Japan and other Asian countries, ingestion of seaweed soup is common in the daily diet [11]. The average intake of iodine from kelp in Japan was estimated to be 1000–3000 µg/d [7,12]. Even with this high iodine intake from kelp, the incidence of iodine-induced health problems does not appear to be severely elevated in Japan [7,13]. Studies in humans [14,15] have shown that the bioavailability of iodine is lower from seaweed than from pure mineral iodine, based on urinary iodine concentrations (UIC). The iodine bioavailability also varied with iodine status in the participants, the species of seaweed used, and the forms of iodine (organic or inorganic) [14].

Harvest and consumption of seaweed and macroalgae are increasing in Europe and Western countries. The European Food Safety Authority (EFSA) has requested more knowledge on the safety issues of seaweed and macroalgae consumption regarding iodine and other metals present. The concentrations of heavy metals in edible seaweed are generally below toxic levels; however, levels of arsenic (As), cadmium (Cd), and copper (Cu) may exceed toxic levels [16].

The aim of this study is to investigate the bioavailability of iodine, the apparent uptake of metals from sugar kelp, as well as any effects on the thyroid hormone system and liver metabolism. We included two doses (0.5% and 5%) of farmed sugar kelp (*Saccharina latissima*) or potassium iodide (KI) in the diets of female Wistar rats for 13 weeks. As a well-established reference for iodine bioavailability, potassium iodide was included and used to compare and interpret the iodine bioavailability from sugar kelp. This study followed the EFSA guidance on a repeated-dose 90-day oral toxicity study on whole food/feed in rodents [17]. Urinary and faecal iodine were quantified. Tissue concentrations of total As, inorganic As (iAS), and Cd were measured, adding to general health parameters like body weight, feed intake, haematology, and clinical biochemistry. Plasma levels of TSH, T3, fT3, T4, and fT4 levels were measured to monitor iodine status, adding to other thyroid-related hormones and metabolites. Finally, liver metabolites were investigated to determine whether the iodine level and iodine source affected liver metabolism.

## 2. Materials and Methods

### 2.1. Experimental Design

The animal experiment was performed following the ‘EFSA guidance on repeated-dose 90-day oral toxicity study on whole food/feed in rodents’ [17]. The Wistar rat was chosen as a model, as this strain is resistant to developing autoimmune thyroiditis [18], making it a suitable strain for studying bioavailability and thyroid function at high iodine intake. Fifty female Wistar IGS rats from Charles River Laboratories (Denmark) were randomly divided into five experimental groups to compare the bioavailability of iodine and investigate the accumulation of metals from farmed sugar kelp (*Saccharina latissima*). The rats were housed pairwise (*n* = 5), as recommended by EFSA, [17] at 20–22 ± 1 °C with a standard 12/12 h light/dark cycle. All rats were fed their experimental diets ad libitum for 13 weeks after one-week acclimatisation. After 11 weeks on the experimental diets, all rats were housed individually in metabolic cages for 24 h urine and faecal collection. After 13 weeks, the rats were anesthetised using isoflurane (Isoba-vet, Schering-Plow, Denmark) and sacrificed by cardiac puncture. Blood samples were collected, and the liver, heart, and kidneys were dissected and weighed. All tissue and plasma samples were snap-frozen and stored at –80 °C until further analyses. The animal experiment was performed following the approval given by the Norwegian Animal Research Authority (FOTS ID: 14040). No adverse effects were observed in the animals during the experiment.

### 2.2. Sugar Kelp

The farmed sugar kelp (*Saccharina latissimi*) was produced and provided by Lerøy Seafood Group (Bergen, Norway) and contained 3971 mg iodine/kg after freeze-drying. Inorganic As (350 µg/kg), total As (58 mg/kg), Cd (0.45 mg/kg), and Cu (2.1 mg/kg) were also quantified in the freeze-dried sugar kelp. The sugar kelp was freeze-dried and homogenised before adding it to the experimental diets. The sugar kelp was included at two dietary levels, with the lowest level (0.5% *ww*) contributing to an iodine intake above the anticipated human intake and the highest level (5% *ww*), which is tenfold higher levels than recommended human intake of iodine. To avoid nutritional imbalance and potential metabolic disturbance, EFSA recommends no higher than 5% inclusion of sugar kelp [18]. KI (CAS: 7681-11-0, Sigma-Aldrich, MO, USA) was included in the experimental diets in concentrations matching the iodine content in the diets with sugar kelp.

### 2.3. Experimental Diets

Five experimental diets were used in this feeding trial. A regular AIN-93G diet from Ssniff Spezialdiäten GmbH (Soest, Germany) was used for one-week acclimatisation. The other experimental diets were also produced by Ssniff Spezialdiäten GmbH and were based on the macro- and micronutrient compositions of the AIN-93G diet. The iodine content in the experimental diets was increased by adding KI or sugar kelp. The diets containing sugar kelp had 17 and 200 mg iodine/kg diets, in the low- and high-iodine diets, respectively. The corresponding diets based on KI had 14 and 160 mg iodine/kg diets. Table 1 shows the measured iodine, inorganic and total As, Cd, and Cu content in the experimental diets.

### 2.4. Energy Intake, Feed Efficiency and Iodine Bioavailability

The rats were fed three times per week and given approximately 15 g of feed per rat per day. Feed leftovers were collected and weighed. Total energy intake was calculated based on the feed eaten and the energy content in the experimental diets. Feed efficiency was calculated based on body weight gain and energy intake during the 13-week feeding period. Iodine bioavailability was calculated by measuring iodine content in urine and faeces collected for 24 h and the corresponding food intake in week 11 of the experiment.

Urinary and faecal iodine concentrations were determined in the samples using inductively coupled plasma-mass spectrometry (ICP-MS) after dilution with 1% tetramethylammonium hydroxide (TMAH), as described earlier [19]. Urine and faecal samples were measured in duplicates. Sugar kelp and feed samples were added to 1 mL TMAH and 5 mL deionised water before extraction at 90 ± 3 °C for 3 h in a water bath shaker. The samples were then diluted and centrifuged. Before quantification, the samples were filtered through a 0.45 μm single-use syringe and disposal filter. Tellurium was used as an internal standard to correct for instrument drift. Iodine concentration in the samples was determined by ICP-MS. Total As, Cd, and Cu were determined in feed, urine, and faecal samples using ICP-MS, as described earlier [20]. iAs was quantified in feed and liver tissue using an anion-exchange HPLC-ICPMS (1260 HPLC, 7900ICP-MS, Agilent Technologies, Wilmington, DE, USA), as described earlier [20].

### 2.5. Blood Haematology

Blood haematology screening was performed at the termination. Five µL of EDTA-treated full blood was analysed with an Abaxis Vetscan HM5 Haematology Analyser. Plasma levels of glucose, total cholesterol, triacylglycerol, alanine aminotransferase (ALT), aspartate aminotransferase (AST), alkaline phosphatase (ALP), bilirubin and creatinine were measured with an automated MaxMat PL II diagnostic analyser system (MAXMAT S.A., Montpellier, France) using kits from MaxMat (Montpellier, France) and DIALAB GmbH (Neudorf, Austria).

### 2.6. Liver Metabolomic Profiling and Data Analysis

To identify possible effects of excessive iodine intake as KI or sugar kelp on liver metabolism, global metabolite profiles were determined in liver samples. Semi-quantitative metabolomic analysis was performed by Biocrates Metanomics Health GmbH (Metanomics Health GmbH, Germany). Briefly, extracts were separated into lipid and polar fractions before analysis on two platforms: gas chromatography-mass spectrometry (GC-MS; Agilent 6890 GC coupled to an Agilent 5973 MS System, Agilent, Waldbronn, Germany) and liquid chromatography-MS/MS (LC-MS/MS; Agilent 1100 HPLC-System, Agilent, Waldbronn, Germany, coupled to an Applied Biosystems API4000 MS/MS-System, Applied Biosystems, Darmstadt, Germany). Pooled reference samples derived from aliquots of all samples were analysed in parallel, and global profiling data were normalised against the median in the pooled reference samples. Known compounds were identified and compared with Biocrates Metanomics Health library entries of purified standards. All raw data were further processed in Qlucore omics explorer 3.5 (Qlucore AB) and GraphPad Prism v6.1 (GraphPad Software Inc., San Diego, CA, USA).

### 2.7. Plasma Measurements of Iodine-Regulated Thyroid Hormones

TSH, free T3, and free T4 levels were measured in plasma samples from termination and were performed with (rodent-based) ELISA kit catalog number KA2336, KA4012, and KA4013 from Abnova (Abnova Corporation), as described in the manufacturer’s instructions. Additionally, plasma metabolites thyroxine (T4), 3,3′,5-triiodothyronine (T3), 3,3′,5′-triiodothyronine (rT3), 3,5-diiodothyronine (3,5-T2), 3,3-diiodothyronine (3,3-T2), 3-iodothyronine (T1), thyronine (T0), 3-iodothyroacetic acid (T1Ac), 3,5-Diiodothyroacetic acid (Diac), triiodothyroacetic acid (Triac) and tetraiodothyroacetic acid (Tetrac) were analysed by isotope-dilution mass spectrometry modified from our previous study [21]. Briefly, 25 µL plasma was spiked with isotopic-labelled (^13^C)-thyroid hormone standards (cT2, cT3 and cT4) and after antioxidant treatment (0.1 mL, 25 mg/mL ascorbic acid, R,R-dithiothreitol and citric acid solution) and protein denaturation by urea (8 M in 1% NH_4_OH) enriched using solid-phase microextraction (10 mg, HRP, ThermoFisher Scientific, Waltham, MA, USA). Samples were eluted with 150 µL methanol and reconstituted in 100 μL of 5% methanol containing an instrument control standard (crT3). Five microliter of the sample extracts were injected on a nanoflow ultra-high performance liquid chromatography system (ThermoFisher Scientific) with a preconcentration trap-column setup described elsewhere [22,23]. Trapped thyroid hormones were eluted to the analytical column (PepMap RSLC, C18, 2 µm, 100 Å, 75 µm × 250 mm, ThermoFisher Scientific) at 300 nL/minute with mobile phases comprising water, acetonitrile, and 0.1% formic acid. Thyroid hormones and associated metabolites were detected by Orbitrap high-resolution tandem mass spectrometry (nanoLC-HRMS/MS, Q Exactive HF, ThermoFisher Scientific) in parallel-reaction monitoring acquisition mode using an ion inclusion list. Data analysis was conducted in TraceFinder v4.1 (Thermo Scientific Inc.), and for quantification, a six-point calibration curve (0.24–7.8 pmol/mL) run in duplicates was applied.

### 2.8. Statistical Analysis

All the reported results are shown as mean ± SEM and analysed for normal distribution and homogeneity of variance before statistical analyses were conducted. Experimental data were assessed using two-way ANOVA, including only rats given low and high levels of iodine from sugar kelp or KI, using iodine level and iodine source as two independent variables. A significant effect caused by the iodine source is denoted by different letters (a, b), and an effect of the iodine level is marked with # (*p* < 0.05). A tendency, a *p* value between 0.05–0.1, is marked with $. If an interaction effect between iodine level and iodine source was observed, a post hoc Fisher least significant difference (LSD) test was performed. Interaction effects are marked with * and the connected *p* value. All statistical analyses and figures, besides the PCA plot and connected statistics, were performed using GraphPad Prism v6.1 (GraphPad Software Inc., San Diego, CA, USA). PCA plots and heatmaps were generated using Qlucore omics explorer 3.5 (Qlucore AB, Lund, Sweden). Quantitative results below LOQ were not included in the statistical analysis or generated figures.

## 3. Results

### 3.1. Iodine from Sugar Kelp (Saccharina latissima) Has Lower Bioavailability Than Iodine from KI

Rats were fed iodine from two sources, KI and sugar kelp in two doses. The bioavailability of iodine was quantified in week 10 (Figure 1). The iodine intake (Figure 1a), adding to urinary and faecal iodine excretion, was quantified after 24 h in metabolic cages. The bioavailability of iodine was evaluated by quantifying the levels in urine (Figure 1b). Iodine excreted into urine from rats fed KI diets was 95 ± 2.4% and 94 ± 3.4% and from rats fed sugar kelp diet 73 ± 3.0% and 81 ± 2.6%. Rats fed sugar kelp had lower iodine excretion in urine than rats given KI (*p* < 0.05). An iodine bioavailability of 73 ± 3.0% and 81 ± 2.6% was measured from rats with low and high sugar kelp intake, respectively. This is relatively high, but still, lower compared to rats fed KI (Figure 1b). Lower iodine excretion in urine from rats given sugar kelp was accompanied by higher iodine levels in faeces. On average, 7.9 ± 1.1% of the total iodine was found in the faeces of rats given kelp in the diet. In contrast, in rats fed diets supplemented with KI, 0.8 ± 0.17% of total iodine was found in faeces, independent of the iodine levels (Figure 1c).

### 3.2. Excretion and Accumulation of Total As and iAs in Rats Fed Diets Containing Sugar Kelp

Compared to terrestrial plants, seaweed is known to contain higher levels of As. In our study, 0.5% or 5% sugar kelps (Table 1) were mixed into the diet, and we calculated total As intake per day based on feed intake and quantified total As exposure and levels in urine and faeces collected after 24 h in metabolic cages (Figure 2a–c). Total As exposure was higher in the group fed 5% than in the group fed 0.5% sugar kelp (Figure 2a). No differences in faecal excretion (%) of As were observed between rats fed 0.5 and 5% kelp in the diet (Figure 2b). However, in rats fed the highest level of kelp, As was also excreted in urine (Figure 2c). In rats fed high-dose kelp, total As excretion in urine was 7.8 ± 1.1%, whereas As excreted in faeces from the low-and high-dose sugar kelp-fed rats were 56.7 ± 6% and 72 ± 8.5%, respectively (Figure 2c). The highest hepatic accumulation of total As was measured in rats given a high-dose kelp in the diet (Figure 2d). Accumulation of total As in kidneys increased in the groups fed kelp (*p* < 0.001), with the highest As levels in the rats fed 5% kelp (Figure 2e). Only the 5% kelp diet had a detectable level of inorganic As (iAs) (Table 1) in the diet, but iAs were still not detectable in the liver from rats fed this diet.)

### 3.3. Excretion and Accumulation of Cd and Cu from Sugar Kelp

The Cd level in the sugar kelp used in the experiment was 0.45 mg/kg dry matter. The estimated dietary Cd intake from the two sugar kelp diets was 0.1 and 0.39 µg per day (Figure 3a). Relative high levels of Cd were measured in faecal samples in rats fed 0.5 and 5% kelp, and 97.1 ± 2.3% (0.5% kelp) and 92.7 ± 3.2% (5% kelp) of the ingested dietary Cd were excreted in faeces within 24 h (Figure 3b). In liver and kidney tissues, Cd levels above LOQ were only observed in rats fed 5% kelp in the diet (Figure 3c,d).

The Cu concentrations in the experimental diets with 0.5 or 5% sugar kelps were comparable with the other experimental diets (Table 1). Hence, the Cu concentrations in liver and kidney tissues were not affected by including 0.5 or 5% kelp in the diet.

### 3.4. Levels of Iodine in the Diet Do Not Affect Body Weight Development and Feed Efficiency, but Modulate Energy Intake

Prolonged excessive iodine exposure has been reported to negatively impact thyroid glands and thyroid hormones. Since thyroid hormones affect metabolism and regulate energy balance, body weight development and energy intake were measured, and feed efficiency calculated. No difference in body weight development was observed in rats fed low and high iodine, from KI or kelp, during the experiment (Figure 4a). Despite no differences in body weight and feed efficiency, a tendency (*p* = 0.09) to increased energy intake was observed in rats fed the low iodine diets compared to the high iodine diets (Figure 4b,c). Heart and kidney masses were unchanged in the low-and high-iodine-fed rats (Figure 4d,e). At termination, increased liver masses were observed in rats fed KI, compared to rats fed sugar kelp (Figure 4f).

### 3.5. Thyroid-Stimulating Hormone (TSH), Free Triiodothyronine (fT3), Free Thyroxine (fT4), and Other Thyroid-Related Hormones

Plasma concentrations of iodine-regulated thyroid hormones were measured after 13 weeks of dietary treatment (Figure 5a,b). TSH and Free T3 levels in plasma were not different in any of the dietary groups but decreased free T4 levels were evident in plasma from the high iodine-fed rats (KI and kelp) (Figure 5a). 3,5-Diiodo-L-thyronine (T2), with a potential suppressing TSH effect, was increased in rats fed kelp, but no differences were demonstrated in reverse T3 (rT3) levels (Figure 5b). A tendency (*p* value = 0.07) towards increased T3 and reduced T4 (*p* value = 0.07 and *p* value = 0.08, respectively) was observed in rats fed the highest doses of iodine, independent of the iodine source (Figure 5b).

### 3.6. Blood Haematology and Plasma Markers for Kidney Function and Liver Damage

Excessive iodine intake or iodine deficiency has earlier been suggested to affect several physiological functions, including white and red blood cell count, plasma triacylglycerol, and cholesterol, alongside kidney and liver functions. White and red blood cell counts alongside lymphocytes were unchanged in the rats after 13 weeks on these experimental diets (Figure 6a). No differences in plasma blood glucose, triacylglycerol, or cholesterol were observed in any of the groups (Figure 6b). Plasma creatinine levels were quantified to evaluate kidney function, however, no increase in plasma creatinine was detected in any of the groups (Figure 6c). Standardised markers for detecting liver damage were measured in plasma samples, and no changes in alanine aminotransferase (ALT), aspartate aminotransferase (AST), or alkaline phosphatase (ALP) were detected in the groups fed high levels of iodine, independent of iodine level and source. However, a tendency towards a reduction in plasma bilirubin (*p* value = 0.061) was observed in rats fed the high iodine diets compared to rats fed the low iodine diets independent of the iodine source (Figure 6d).

### 3.7. Rats Fed a High Iodine Diet Based on Kelp Have a Distinct Hepatic Metabolite Profile

Rats fed the highest levels of sugar kelp (5%) in the diet had a distinct metabolite profile in the liver, and principal component analysis (PCA) revealed a separation of the measured liver metabolites in rats fed the highest dose of sugar kelp from the other dietary groups (Figure 7a). Fourteen metabolites were significantly regulated when comparing all groups, including metabolites involved in cholesterol synthesis and redox potential, eicosaenoic acid and several triacylglycerols (Figure 7a). Comparing liver metabolomics data from animals fed the two sources of iodine (KI and kelp) showed that six metabolites were significantly altered (Figure 7b), and three of these have potential antioxidant activity. Gamma-tocopherol and alpha-tocopherol were downregulated in rats fed kelp, but in contrast, an increase in beta-carotene was observed (Figure 7c). A separation in the PCA of measured liver metabolites after extraction of beta-carotene was still evident. In addition to the three metabolites with potential antioxidant activity, Coenzyme Q10 (oxidized) was lower in rats fed the highest dose of kelp, and Campesterol and Beta-sitosterol were significantly lower in rats fed kelp, independent of dose (Appendix A). Comparing the low and high iodine groups, independent of iodine source, revealed 13 differently affected metabolites in the liver, and eight of these metabolites were different triacylglycerols, with lower levels in the high iodine-fed rats (Appendix A).

## 4. Discussion

Here, we aimed to compare the bioavailability of iodine in sugar kelp with KI in female Wistar rats. The experiment further aimed to uncover the potentially harmful effects of prolonged kelp intake and a high iodine intake on liver metabolism and thyroid function, as well as general health parameters like body weight, feed intake, haematology, and clinical biochemistry.

We demonstrated higher faecal iodine levels and significantly lower iodine bioavailability in rats fed iodine from sugar kelp than from KI. This follows earlier human and animal studies indicating that iodine bioavailability from seaweed in general is lower than that from KI [14,15,16,24]. However, iodine bioavailability in different types of seaweed varies, and overall bioavailability estimates are significantly influenced by the iodine status of the study population [14]. A recent systematic review reported that the bioavailability of iodine from brown algae in vivo varied from 31% to 90% [25]. The bioavailability of iodine and iodine absorption from different seaweeds has been proposed to be affected by the polysaccharide matrix of the seaweed [15] and may delay iodine absorption compared with KI. Another factor affecting bioavailability is the form of iodine in seaweed. Seaweed contains both organic and inorganic iodine [26], and most brown algae and algae with a high iodine content contain most of the iodine in the form of iodide [25]. The exact mechanisms underlying the lower iodine bioavailability in sugar kelp compared to KI found in this study were not investigated, but differences in matrix, fibre content, and excretion route could be possible explanations. Urinary iodine excretion is one of the most reliable methods for determining iodine excretion and bioavailability. The major source of error is caused by incorrect or incomplete sample collection. Another potential source of error is the iodine measurement itself and other preanalytical variables. Still, an iodine bioavailability of 73–81% from sugar kelp is relatively high, and seaweed consumption could potentially be a good supplement for improving iodine status in iodine-deficient populations. The significantly lower bioavailability of iodine from sugar kelp demonstrated in this study is of importance for the consumer with regard to the amount of sugar kelp possible to eat without exceeding the tolerable upper intake level of iodine.

No effect of kelp or high iodine levels was seen regarding body weight development, feed intake, or plasma markers of liver- or kidney function. To evaluate the effects of high iodine intake, we investigated thyroid hormonal changes related to the very high dietary intake of iodine. We analysed various thyroid-related metabolites. However, few changes were observed in any of the groups treated with either KI or kelp, independent of doses. The acetylated metabolites, alongside T1 and T0, were below the LOQ, and only minor changes were observed in TSH, T2, T3, T4, and fT4. This agrees with another study by Yoshida et al. where no changes were observed in TSH, T3, and T4 in rats. Notably, the iodine levels in our experimental diets were slightly lower than the doses used by Yoshida et al. [24]. However, Gao et al. [27] observed hyper-and hypothyroidism in rats when fed chronic high levels of iodine in the diet. The observed changes were seen after four and six months of intake. Collectively, our results support the earlier observed high tolerance of iodine in rats [24,28] and highlight that exposure time, and potentially the experimental background diet, could be more important than iodine doses concerning hormonal changes, at least in rats.

Seaweed contains dietary starch, fibre, and small amounts of omega-3 fatty acids [29], adding to several micronutrients, metals [30], and low levels of organochlorines [31]. Total As and iAs levels in seaweed have been of concern, and high levels of iAs have been observed in consumers of Hijiki seaweed [32]. The levels of iAs in the farmed sugar kelp used in this experiment were low and barely above LOQ in the analysed feed, and detectable levels were not observed in liver tissue from rats fed 0.5 or 5% sugar kelp in the diet. Considering the low levels of iAs in the experimental diets and that rodents metabolize iAs much quicker than humans this was not surprising [33]. However, despite elimination of more than 65% of the total As intake from kelp, accumulation of total As in both liver and kidney tissue was observed at termination. Despite accumulation of total As in the liver of rats fed kelp, no significant increase in liver weight or markers for liver damage measured in plasma indicated any severe liver damage after 13 weeks of kelp intake. Following this, liver metabolomic data do not indicate any negative effects of consuming kelp. Despite no effects of kelp intake on plasma markers for liver damage or function, we observed changes in the levels of antioxidants in the liver in these rats. The physiological effects of the increased beta-carotene and reduced gamma and alpha-tocopherol levels in the liver of these animals are not further evaluated in this study.

Cd is present in seaweed and may negatively affect the kidneys and induce conditions of liver failure [34]. Surprisingly, the highest accumulation of Cd was observed in the liver tissue, but the levels were only marginally above LOQ. No kidney damage or impairment in kidney function was observed based on kidney markers measured in plasma. Dietary Cd has earlier been shown to have a low absorption rate in the gut, where data from humans demonstrate that approximately 3–5% are absorbed, while the rest passes through the gastrointestinal tract unchanged [35]. Data from our study demonstrate that more than 92% of total dietary cadmium is not absorbed and is found in faeces. Despite the relatively high faecal levels of Cd in rats fed kelp, Cd accumulation in both the liver and kidney was higher in the kelp-fed groups. Based on the level, route and total time of exposure used in earlier experiments, it is unlikely that Cd exposure from kelp in this experiment has any harmful or toxic effect on the liver in these rats [35].

Liver metabolomic analyses were performed to identify effects of high iodine and kelp inclusion in the diet. The data from liver metabolomics demonstrated that the iodine levels irrespective of the source modulated metabolites in the liver, and rats fed the highest dose of sugar kelp further differed from the other dietary intervention groups. The separation in liver metabolites from rats fed sugar kelp and the other experimental diets was mainly driven by differences in levels of antioxidants. Excessive iodine intake through drinking water has earlier been reported to induce hepatic steatosis through altered thyroid hormone metabolism involving increased oxidative stress in mice [36]. However, in contrast to us, they observed significant differences in TSH and T3. Our results demonstrate a significant decrease in several clusters of TAGs in rats fed high iodine diets independent of the iodine source. Evaluating liver weight, no differences were observed when comparing rats fed low and high iodine doses, but lower liver weight in rats fed kelp compared to KI was observed, independent of iodine dose. However, Kumar et al. (2015) did not observe a reduction in liver weight when supplementing a high fat diet with the green seaweeds *Ulva ohnoi* or *Derbesia tenuissima*, but reduced enlargement of hepatic fat vacuoles was observed in rats fed *Ulva ohnoi* [29].

Earlier studies have reported that individuals with low antioxidant levels in the liver are potentially more vulnerable to liver damage later in life [37]. Liver metabolomics revealed that five percent sugar kelp in the diet affected the hepatic redox potential. The high kelp group had lower alfa- and gamma-tocopherol (vitamin E), but higher beta-carotene (provitamin A). Both alfa- and gamma-tocopherol play a vital role in protecting against oxidative stress and lipid peroxidation and in the maintenance of tissue homeostasis. Inorganic arsenic can negatively affect alpha-tocopherol levels [38], however, this effect has not been observed with organic arsenic. Iodine is a redox-sensitive element forming various organic and inorganic compounds, which potentially affect the redox system [39]. The properties of the coenzymes Q9 and Q10 are related to their redox state and are the only lipophilic antioxidants synthesised in humans [40], and Q9 and Q10 are useful markers for oxidative stress [41,42]. Q10 can react with the reduced form of alfa-tocopherol, producing less reactive pro-oxidants or oxidise itself to reverse the oxidised form of alfa-tocopherol. In inorganic arsenic-exposed rats, co-administration of ascorbic acid and alpha-tocopherol improved arsenic-induced altered microsomal functions in both liver and kidney tissues [38]. Lower levels of hepatic alpha-tocopherol and Q10 in rats fed the highest levels of sugar kelp could indicate increased oxidative stress. In contrast, beta-carotene levels were significantly increased in the dietary group fed the highest dose of sugar kelp. Most seaweeds and sugar kelp are rich in beta-carotene as a part of photosynthetic pigments [43]. Beta-carotene has a strong antioxidant effect and can prevent NAFLD development [44]. Whether higher levels of beta-carotene may compensate for reduced alpha-and gamma-tocopherol levels in rats fed the highest dose of kelp remains an open question. Further research is needed to clarify if and how sugar kelp affects oxidative stress.

## 5. Conclusions

Overall, our results demonstrated that the bioavailability of iodine from sugar kelp is significantly lower than that from KI. Despite lower iodine bioavailability, more than 73% of ingested iodine from sugar kelp was absorbed. A lower bioavailability of iodine from sugar kelp has an impact on the maximal amount of sugar kelp possible to consume before exceeding the tolerable upper intake level. Excessive iodine intake from kelp or KI did not affect TSH, fT3, and rT3. However, a small reduction in fT4 and T4 levels was observed in rats fed the highest doses of iodine. Collectively, these results confirmed a high tolerance for excessive iodine intake in healthy rats and that an exposure period of 13 weeks did not strongly affect important thyroid hormones. Sixty percent of total As and 90% cadmium intake from sugar kelp were eliminated in faeces. Intake of sugar kelp over 13 weeks did not affect plasma markers of liver or kidney damage. Liver metabolomic data revealed a reduction in alfa-and gamma-tocopherol and an increase in beta-carotene levels in rats fed the highest dose of sugar kelp (5%). The potential long-term effects of these changes are unclear. More human intervention studies with well-defined health-related endpoints are necessary to confirm if and how chronic consumption of seaweed and kelp affects health.

## Figures and Tables

**Figure 1 foods-11-03943-f001:**
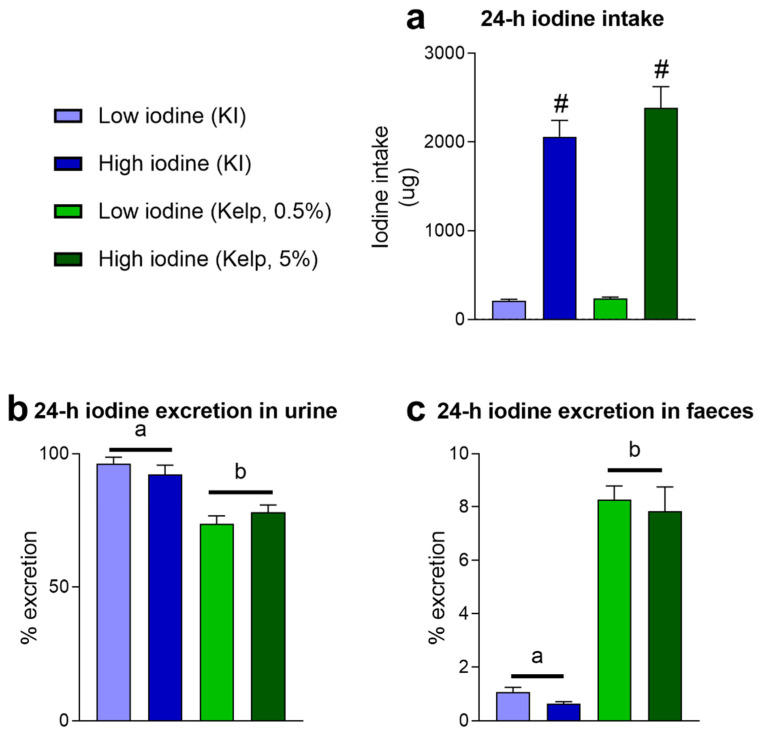
(**a**) 24 h dietary iodine intake and iodine excretion in (**b**) urine and (**c**) faeces in rats fed KI and sugar kelp in week 10. KI and sugar kelp were added to the experimental diets in two doses, low and high doses. Data are presented as mean ± SEM (*n* = 8–10). Different letters (a,b) in the graph denote significant differences (*p* < 0.05) between the iodine sources (kelp, KI), where # indicates a significant difference between the iodine doses (independent of source). Data were analysed using a standard two-way ANOVA in GraphPad Prism.

**Figure 2 foods-11-03943-f002:**
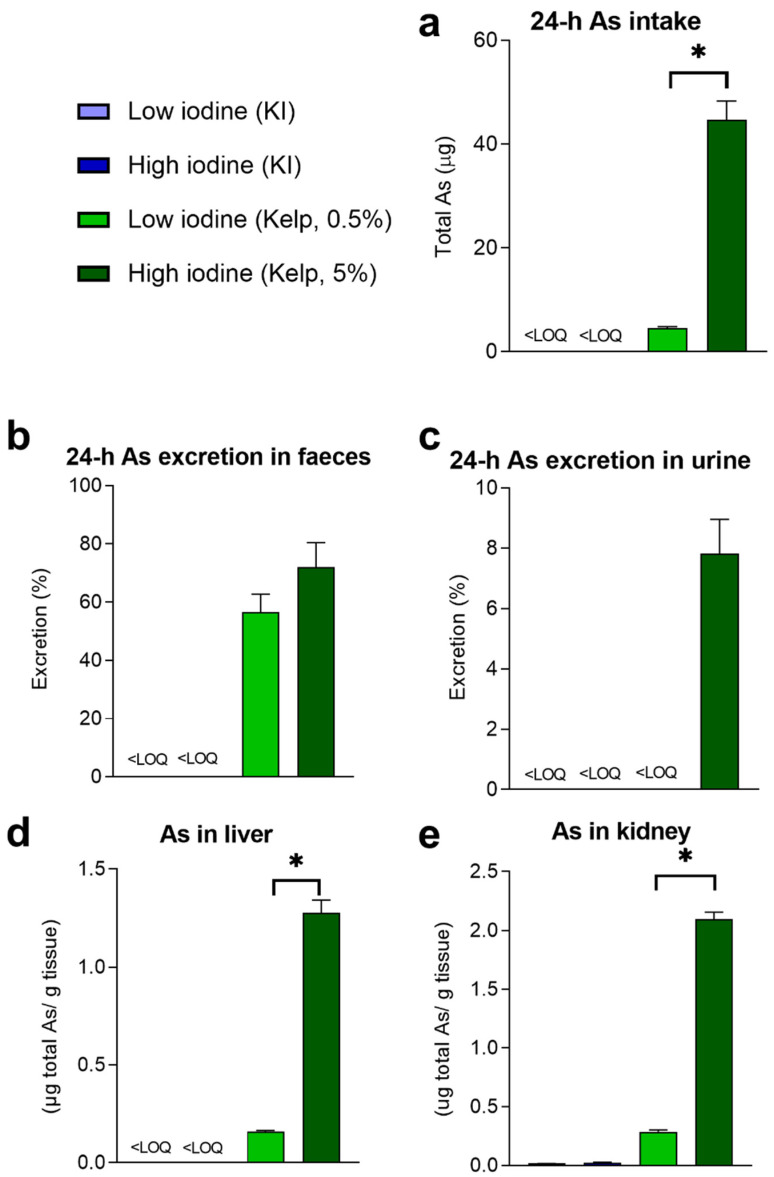
(**a**) 24 h total As dietary exposure and percent excretion in (**b**) faeces and (**c**) urine in rats fed KI and sugar kelp in week 10. (**d**) The accumulated concentration of total As in liver and (**e**) kidney tissue after 13 weeks of feeding. Data are shown as mean ± SEM (*n* = 8–10). Measured values below the limit of quantification (LOQ)_As_ = 0.001 (mg/kg) are marked < LOQ. Significant differences between groups fed the two levels of kelp in the diet are marked with *. Data were analysed with a standard *t*-test.

**Figure 3 foods-11-03943-f003:**
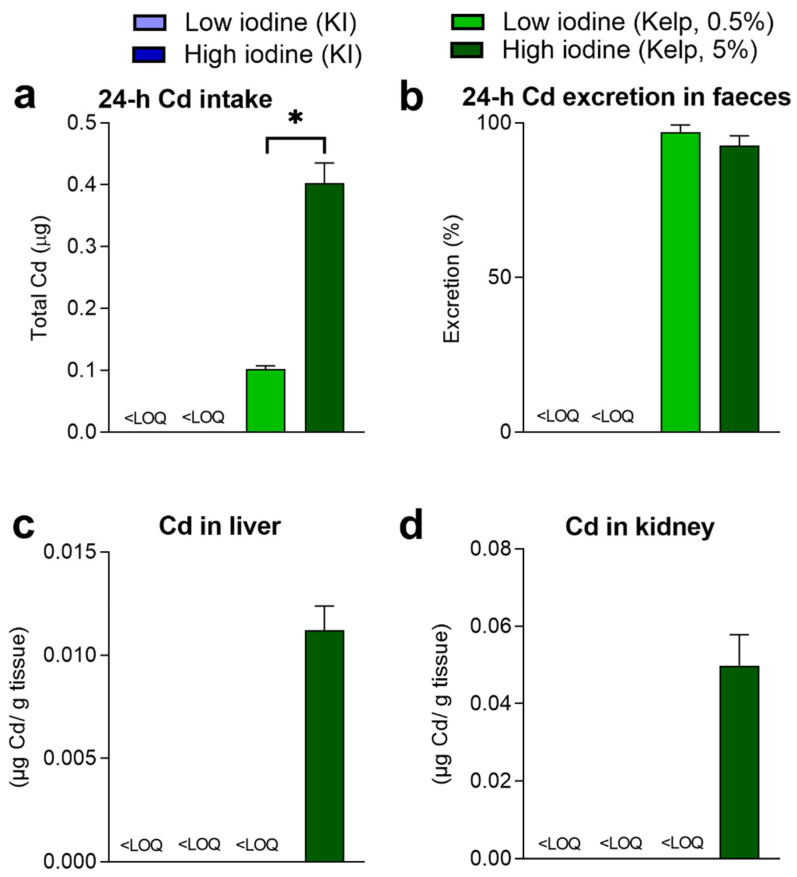
(**a**) 24 h cadmium (Cd) dietary exposure and (**b**) excretion in faeces in rats fed KI and sugar kelp in week 11. (**c**) Accumulated concentrations of Cd in liver and (**d**) kidney tissues after 11 weeks of feeding. Data are shown as the mean ± SEM (*n* = 9–10). Measured values below the limit of quantification (LOQ)_Cd_ = 0.0001 (mg/kg) are marked < LOQ. Significant differences between groups fed the two levels of kelp in the diet are marked with *. Data were analysed with a standard *t*-test; values in rats fed KI were below LOQ.

**Figure 4 foods-11-03943-f004:**
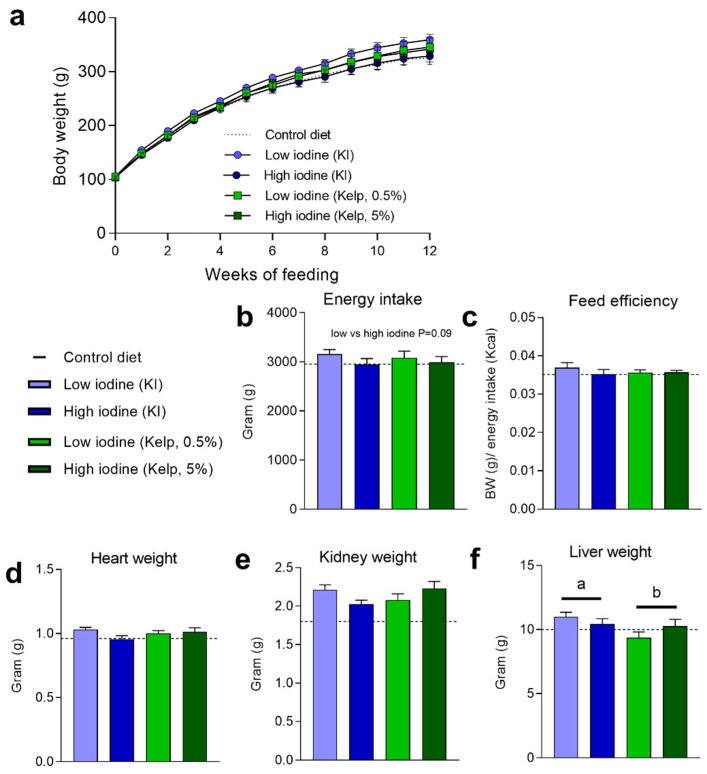
(**a**) Body weight development, (**b**) energy intake, (**c**) feed efficiency, (**d**) weight of heart, (**e**) kidney and (**f**) liver after 13 weeks of dietary treatment. Data are presented as mean ± SEM (*n* = 5). Different letters (a,b) in the graph denote significant differences (*p* < 0.05) between the iodine sources (kelp, KI). Data were analysed by a standard two-way ANOVA in GraphPad Prism.

**Figure 5 foods-11-03943-f005:**
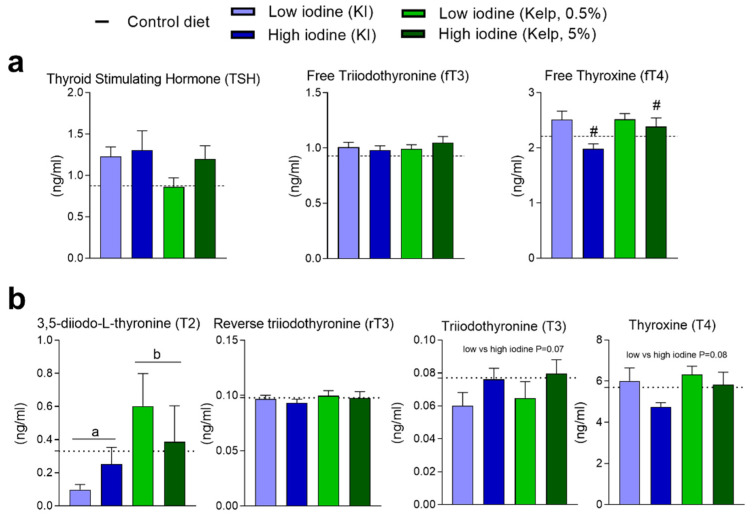
Plasma levels of thyroid hormones after 13 weeks of dietary treatment. (**a**) TSH, fT3, and fT4 analysed with ELISA. (**b**) 3,5-Diiodo-L-thyronine (T2), rT3, T3 and T4 analysed with nanoLC-HRMS/MS. Data are presented as mean ± SEM (*n* = 6–10). Different letters (a,b) in the graph denote significant differences (*p* < 0.05) between the iodine sources (kelp, KI), with # indicating significant differences between the iodine doses (independent of source). Data were analysed using a standard two-way ANOVA in GraphPad Prism.

**Figure 6 foods-11-03943-f006:**
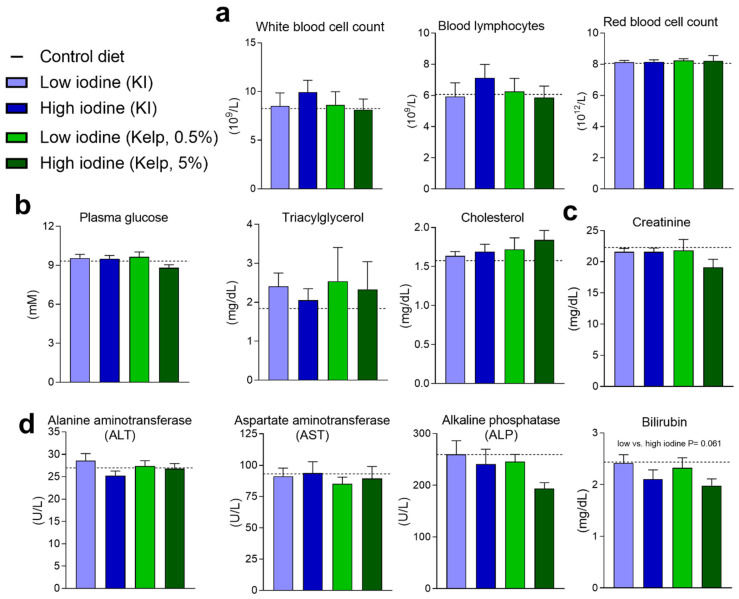
Blood haematology and plasma measurements: (**a**) White blood cells, blood lymphocytes, and red blood cells measured after 13 weeks of dietary treatment. (**b**) Plasma measurements of glucose, triacylglycerol and cholesterol, (**c**) creatinine and (**d**) alanine aminotransferase, aspartate aminotransferase, alkaline phosphatase, and bilirubin. Data are presented as mean ± SEM (*n* = 8–10). Data were analysed using a standard two-way ANOVA in GraphPad Prism.

**Figure 7 foods-11-03943-f007:**
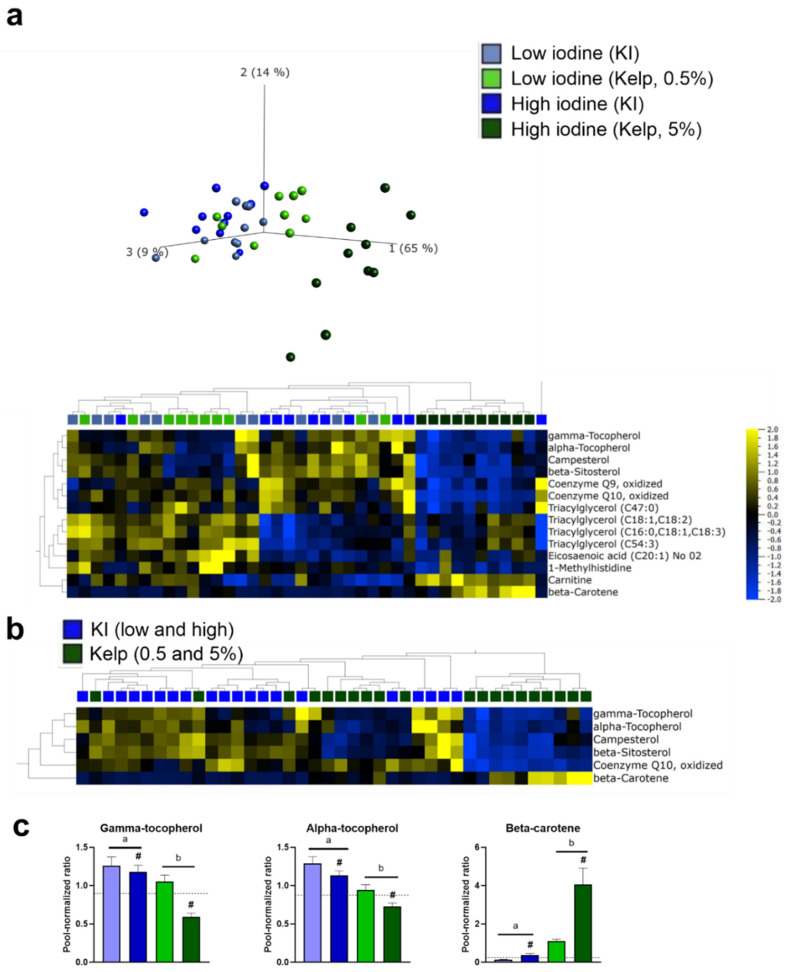
(**a**) PCA including 681 liver metabolites measured in liver tissue harvested after 13 weeks of dietary treatment. Heatmap and hierarchical clustering of liver metabolites differently regulated between iodine sources and doses (q-value < 0.05) (**b**) Heatmap and hierarchical clustering of liver metabolites different between iodine sources (q-value < 0.05). Data were visualised, and statistical tests were performed using Qlucore Omics Explorer. (**c**) Selected metabolites with potential antioxidant activity regulated in rats fed low and high KI and sugar kelp diets. Measured levels were semi-quantitative and normalised against a reference pool sample (*n* = 9–10). Different letters (a,b) in the graph denote significant differences (*p* < 0.05) between the iodine sources (kelp, KI), where # indicates a significant difference between the iodine doses (independent of source).

**Table 1 foods-11-03943-t001:** Concentration of iodine, iAs, total As, Cd and Cu in the experimental diets. All diets were measured in duplicate, and the values are given as mean ± SD.

Diet	AIN-93G Control Diet	Low Iodine (KI)	High Iodine (KI)	Low Iodine (Kelp, 0.5%)	High Iodine (Kelp, 5%)
Iodine (mg/kg diet)	0.15 ± 0.01	14 ± 1.5	160 ± 12	17 ± 1	200 ± 12
Inorganic As (µg/kg diet)	<LOQ	<LOQ	<LOQ	<LOQ	18.8 ± 0.9
Total As (mg/kg diet)	<LOQ	<LOQ	<LOQ	0.27 ± 0.015	3 ± 0.3
Cd (mg/kg diet)	<LOQ	<LOQ	<LOQ	0.006 ± 0.0005	0.027 ± 0.015
Cu (mg/kg diet)	8.7 ± 0.4	9.1 ± 0.1	9.2 ± 0.3	9.4 ± 0.5	9.6 ± 0.2

LOQ_Iodine_ = 0.04 (µg/kg); LOQ_iAS_ = 7.3 (µg/kg); LOQ_As_ = 0.001 (mg/kg); LOQ_Cd_ = 0.0001 (mg/kg); LOQ_Cu_ = 0.004 (mg/kg).

## Data Availability

Data is contained within the article or Appendix A.

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
