# Peer review of "Iodine Bioavailability and Accumulation of Arsenic and Cadmium in Rats Fed Sugar Kelp (*Saccharina latissima*)"

_foods, 2022, doi:10.3390/foods11243943_

Round 1
Reviewer 1 Report
1. The implications of this paper are not clear. There is a lack of content as to whether KI diet or sugar kelp diet is or both mean irrelevant.
2. Why is the period of collecting urine and feces different from that of collecting blood and organs? It is questionable whether it is the right analysis to compare those data.
3. It is described that Q10 reacts when alpha-tocopherol reduced at the last paragraph of “Discussion” and according to the supply data, the Q10 content also decreases. There is a lack of interpretation of these data.
4. In figure caption, meaning of 'a' and 'b' at the top of the bar graph should be described in more detail (ex. a p < 0.05)
Author Response
1) In the revised manuscript we have added sentences in both the discussion and conclusion part of the paper to highlight the possible implications of this paper. The therapeutic range of iodine is small, where levels both below and above this range affect thyroid functions and may have multiple adverse effects in humans. A better estimate of the bioavailability of iodine from sugar kelp has implications on the amount of sugar kelp that can be consumed before reaching the tolerable upper intake level of iodine. Comparing the results to the well-established/used reference potassium iodide makes it possible to interpret and compare our results with results from other studies. We have added a sentence explaining that we wanted to compare the bioavailability of sugar kelp to the well-established potassium iodide (KI).
2) The 24-hour urine and feces collection have to be performed in living animals and within a certain time range before the animals are terminated. Accurate 24-hour urine and feces collection must be performed in metabolic cages, and these metabolic cages induce stress for the animals. Thus, it is necessary to have a wash-out period before termination where the blood and organs are collected. Stress induces several physiological changes in the animals, further affecting measurements such as plasma analysis and the metabolic profile in liver tissue. Considering the relatively long duration period of the experiment, the two weeks from urine and feces collection to termination is considered acceptable with regards the to impact on iodine excretion and accumulated levels of arsenic and cadmium in the experimental animals.
3) A paragraph evaluating the potential effects from the decrease in hepatic Q10 levels is added to the discussion part of the manuscript.
Rats fed a high dose of sugar kelp have reduced levels of alpha-tocopherol and oxidized Q10, indicating potential increased oxidative stress upon sugar kelp intake. However, results from plasma measurements (Figure 6) do not demonstrate liver damage and thereby do not support evidence for increased oxidative stress. In addition, levels of the antioxidant beta-carotene are substantially higher in sugar kelp-fed rats, which also may reduce oxidative stress. In summary, it is difficult to conclude whether sugar kelp induces or reduces oxidative stress in rats, and further research is needed to clarify if and how sugar kelp affects oxidative stress.
4) Additional information about the meaning of different letters at the top of the bar graphs indicating significant differences has been added to each figure caption.
Reviewer 2 Report
The current study is highly important due to increased consumption and use of seaweeds in food and feed industries. It is well written and structured.
Major concerns: Though the manuscript is well written and structured, and relevant experiments are presented, the urinary bioavailability is not always an accurate substitute for the plasma level data, relationship between plasma concentration and renal clearance should be established.
1) Why only urine and faeces have been analysed for iodine? When blood has been collected in the experiment.
2) The faecal excretion of iodine has been on average 7.9% for rats given kelp in the diet with urinary excretion of 73% and 81%, that means that around 12-19% is not accounted for? Without blood/plasma data it is not possible to say whether app. 92% of iodine has been absorbed, and if 73 - 81% has been excreted into urine then 12-19% is then distributed throughout the body? This important aspect has not been discussed. Please add in the discussion pitfall/weakness/uncertainty of quantifying iodine bioavailability by using urinary excretion data.
3) The faecal excretion of iodine has been on average 0.8% for rats given KI in the diet with urinary excretion of 95% and 94%, that means that around 4-5% is not accounted for? Again, without blood/plasma data it is not possible to say whether app. 99% of iodine has been absorbed, and if 95% and 94% has been excreted into urine then only 4-5% is then distributed throughout the body? The percentage of how much iodine can be accounted for is higher for KI diet compared to kelp diet. Is iodine then more bioavailable for absorption into tissue, and excreted into urine more slowly? That should also be discussed in the discussion section.
Results
Lines 215 to 220 are not well formulated and should be rewritten.
For example: The bioavailability of iodine… in urine (Fig 1b). Iodine excreted into urine from rat fed KI diets was 95… and 94…% and rats fed kelp diet 73… and 81…%. Rats fed sugar kelp had lower iodine excretion in urine than rats given KI.
Minor concerns:
Material and methods
This section needs reorganization and better division into sections with appropriate titles:
Chemicals
Experimental design and experimental diets
Sugar kelp and content of iodine…
ICP-MS analyses
Analyses of blood and lever markers
Metabolomics
Statistics
Author Response
1) Urinary iodine concentration (UIC) is considered one of the recommended methods for assessing the iodine status in populations. Most of the dietary iodine intake is excreted in the urine, and the total urinary iodine excretion over 24 hours is therefore a good indicator of iodine intake. To make an overall estimation of how much iodine that is found in plasma, several plasma measurements have to be collected and analyzed. Another potential problem with several iodine measurements in plasma samples is the amount of plasma needed to perform the measurement. This could potentially be done in human studies, but in rats, it could be problematic to get enough blood samples. The plasma samples collected at termination were used to measure levels of thyroid-related hormones and hormones affected by dietary intake of iodine.
2) Several factors could potentially influence the iodine measurements and the quantification of iodine bioavailability. A possible explanation for the gap in iodine intake and excretion (both urine and feces) is the excretion time. Combet et al. (2014) showed a peak in urine excretion early after intake, 2-4 h, however, relatively high urine levels were observed 24 h after intake. Excretion through feces could probably take a longer time, so in an optimal situation, we would collect feces for longer than 24 h to see if the overall recovery of the iodine would increase further. In addition, we have analytical variation and possible sources of error in measuring feed intake (and overestimating the iodine intake) and in the collection of feces and urine, which could influence the overall estimations. We have added a section in the discussion regarding weaknesses and possible errors regarding quantifying iodine bioavailability.
3) We agree that blood samples collected over a longer time period could potentially give additional information about the absorption rate of iodine. Potential preanalytical variables and other sources of error are also discussed in point 2.
*The sentence in the result section is rephrased to; “The bioavailability of iodine was evaluated by quantifying the levels in urine (Fig 1b). Iodine excreted into urine from rats fed KI diets were 95 ± 2.4% and 94 ± 3.4% and rats fed kelp diet 73 ± 3.0% and 81 ± 2.6%. Rats fed sugar kelp had lower iodine excretion in urine than rats given KI (P<0.05).”
Minor:
*The material and method section is reorganized and modified to include subheadings. Hopefully, the material and method sections are more clear and more intuitive now.
Round 2
Reviewer 1 Report
The author diligently revised the corrections pointed out by the reviewers, and the revised manuscript has been improved to be published in this journal.
Therefore, I would like to accept the publication of this paper.
Author Response
Dear reviewer,
Thank you for the possibility of revising the manuscript. We have carefully read through the manuscript for fine/minor spell check and checked the introduction and the relevance of the included references to the research.
Our changes are shown with the track changes function in the attached PDF-file.
Best regards
Even Fjære
